# Effects of Dietary Fishmeal Replacement by Poultry By-Product Meal and Hydrolyzed Feather Meal on Liver and Intestinal Histomorphology and on Intestinal Microbiota of Gilthead Seabream (*Sparus aurata*)

Pier Psofakis [1], Alexandra Meziti [1], Panagiotis Berillis [1,*], Eleni Mente [1,2], Konstantinos A. Kormas [1] and Ioannis T. Karapanagiotidis [1,*]

1 Department of Ichthyology and Aquatic Environment, School of Agricultural Sciences, University of Thessaly, Fytoko Street, 38446 Volos, Greece; psofakis@uth.gr (P.P.); ameziti@gmail.com (A.M.); emente@uth.gr (E.M.); kkormas@uth.gr (K.A.K.)
2 Laboratory of Ichthyology—Culture and Pathology of Aquatic Animals, School of Veterinary Medicine, Aristotle University of Thessaloniki, University Campus, 54006 Thessaloniki, Greece
* Correspondence: pveril@uth.gr (P.B.); ikarapan@uth.gr (I.T.K.)



**Featured Application:** Poultry by-product meal and hydrolyzed feather meal can successfully replace fishmeal at low dietary levels in feeds for farmed gilthead seabream and thus enhance the environmental and economic sustainability of its production.

**Abstract:** The effects on liver and intestinal histomorphology and on intestinal microbiota in gilthead seabream (*Sparus aurata*) fed diets that contained poultry by-product meal (PBM) and hydrolyzed feather meal (HFM) as fishmeal replacements were studied. Fish fed on a series of isonitrogenous and isoenergetic diets, where fishmeal protein of the control diet (FM diet) was replaced by either PBM or by HFM at 25%, 50% and 100% without amino acid supplementation (PBM25, PBM50, PBM100, HFM25, HFM50 and HFM100 diets) or supplemented with lysine and methionine (PBM25+, PBM50+, HFM25+ and HFM50+ diets). The use of PBM and HFM at 25% fishmeal replacement generated a similar hepatic histomorphology to FM-fed fish, indicating that both land animal proteins are highly digestible at low FM replacement levels. However, 50% and 100% FM replacement levels by either PBM or HFM resulted in pronounced hepatic alterations in fish with the latter causing more severe degradation of the liver. Dietary amino acid supplementation delivered an improved tissue histology signifying their importance at high FM replacement levels. Intestinal microbiota was dominated by *Proteobacteria* (58.8%) and *Actinobacteria* (32.4%) in all dietary groups, but no specific pattern was observed among them at any taxonomic level. This finding was probably driven by the high inter-individual variability observed.

**Keywords:** nutrition; aquaculture; fishmeal replacement; land animal proteins; histology; intestinal microbiota; *Sparus aurata*

## 1. Introduction

Gilthead seabream (*Sparus aurata*) is one of the most important carnivorous farmed fish species in European aquaculture with an annual production of approximately 186,000 mt [1]. As aquaculture is becoming the major fish-food production sector [2] there is a search for suitable protein sources in aquafeeds that are alternatives to fishmeal to enhance its environmental and economic sustainability. Fishmeal was, and in many cases remains the primary protein source for the nutrition of farmed fish. However, it has become necessary to use low fishmeal diets because the global availability of fishmeal is stagnant, especially for those sourced from the wild, and its price has increased [3]. Land animal proteins, such as hydrolyzed feather meal (HFM) and poultry by-product meal (PBM) are

currently incorporated in European aquafeeds. After their re-approval in 2013, proved to be valuable feedstuffs for dietary fishmeal replacement in the diet of most fish species [4–6], including gilthead seabream [7,8]. Although the poultry sector is responsible for a substantial proportion of greenhouse gases emissions [9], these feedstuffs provide a valuable mean of animal by-products utilization and upgrade the ecological efficiency of the whole poultry production process [10]. Thus, the use of land animal proteins could enhance aquaculture's sustainability and eco-efficiency, as these have a more favorable carbon footprint and a higher environmental efficiency when compared with fishmeal and plant alternatives [11,12].

Dietary protein manipulations, however, are known to affect the functionality of the digestive system [13,14]. A functional digestive system is a prerequisite for the optimal growth of fish with the liver being the main organ of nutrient deposition and metabolism and the intestine being the primary site of nutrient digestion and absorption. Therefore, studying any possible effects and alterations in the histomorphology of these tissues is fundamental for the evaluation of the use of land animal proteins as fishmeal substitutes. Most studies have focused on the effects of plant proteins as fish meal replacements [15] with high substitution levels resulting in marked changes in hepatic and intestinal tissues, such a reduced number of goblet cells, lipid accumulation in hepatocytes, shorter and thinner mucosal folds and villi, steatosis, submucosal layer hypertrophy and impaired structural integrity of the gut [13,15–17]. These alterations are mainly due to the presence of various anti-nutritional factors [18] which in turn cause pathophysiological changes in the gastrointestinal tract and reduce nutrient digestibility.

On the other hand, very little is known about the effects of land animal proteins as fishmeal replacements on the liver and intestinal histology. Findings from the limited studies that have been reported up to date have revealed that high inclusion levels of land animal proteins may induce hepatic steatosis and increase hepatic lipid vacuolization in *Lateolabrax japonicus* [19], in hybrid grouper [20,21] and in *Lates calcarifer* [22]. In addition, negative effects on the intestinal histology have also been reported with the fishmeal replacement by land animal proteins [22,23].

Gut microbes are essential for host nutrition and immunity [24] and changes in their community composition are related to stress and dysbiosis. Fish gut microbes are linked to the diet since different microbiota persist under different nutritional conditions along with the different enzymes produced (proteases, lipases, esterases, cellulases) that contribute to better food digestion by the host [25,26]. It has been shown that the use of alternative protein sources can alter the gut microbiome of the host having a beneficial impact on growth and immunity by triggering lactic acid bacteria (LAB) and cytokines respectively [27].

The present study addressed the effects of poultry by-product meal and hydrolyzed feather meal on liver and intestinal histomorphology and on the intestinal microbiota of gilthead seabream (*Sparus aurata*).

## 2. Materials and Methods

All experimental procedures were conducted according to the guidelines of the EU Directive 2010/63/EU regarding the protection of animals used for scientific purposes. The experiments were performed at the registered experimental facility (EL-43BIO/exp-01) of the Laboratory of Aquaculture, Department of Ichthyology and Aquatic Environment, University of Thessaly by FELASA accredited scientists (functions A–D).

### 2.1. Feeding Trials and Experimental Diets

Two feeding trials were conducted in which the growth data were not the object of the present study and are described in detail elsewhere [7,8]. Briefly, gilthead sea bream (*S. aurata*) juveniles were raised in glass tanks (125 L) with recirculating seawater of standard water quality (21.0 ± 1.0 °C, pH at 8.0 ± 0.4, salinity at 33 ± 0.5 g/L, dissolved oxygen at >6.5 mg/L, total ammonia nitrogen at <0.1 mg/L). In feeding trial I, juveniles

with an initial mean weight of 2.5 ± 0.2 g were raised in quadruplicate groups (25 fish/tank, 4 tanks/dietary group). For 100 days they were fed to satiation with one of the five isonitrogenous (50%) and isoenergetic (21 KJ/Kg) experimental diets [7,8], where the FM protein of the control diet (FM diet) was replaced by either poultry by-product meal (PBM) at 50% (PBM50 diet) and 100% (PBM100 diet) or by hydrolyzed feather meal (HFM) at 50% (HFM50 diet) and 100% (HFM100 diet). In feeding trial II, juveniles with an initial mean weight of 2.9 ± 0.3 g were raised in triplicate groups (25 fish/tank, 3 tanks/dietary group). For 110 days they were fed to satiation with one of the seven isonitrogenous (50%) and isoenergetic (21 KJ/Kg) experimental diets [7,8]. These diets used the same FM control diet as before, but the FM protein was now replaced by PBM and HFM at lower levels: 25% without amino acid supplementation (PBM25 and HFM25 diets), 25% supplemented with lysine and methionine ((PBM25+ and HFM25+ diets), and 50% supplemented with lysine and methionine (PBM50+ and HFM50+ diets). At the end of each feeding trial, fish were weighed individually and euthanized with an overdose of tricaine methanesulfonate (MS 222, 300+ mg/L) according to the Directive 2010/63/EU and FELASA guidelines.

### 2.2. Histological Analysis and Measurements

For the feeding trial I, two fish per tank were randomly sampled (eight fish per dietary group). The liver of each fish was removed quickly and weighed for the determination of hepatosomatic index. Liver and midgut samples were collected from each fish, fixated into 10% formalin in filtered seawater for 24 h at 4 °C and then were immediately dehydrated in graded series of ethanol, immersed in xylol, and embedded in paraffin wax. Intestinal samples from the HFM100 group were not taken because the fish intestine was too thin. A part of the liver of fish was also collected for the determination of its fat content by exhaustive Soxhlet extraction using petroleum ether on a Soxtherm Multistat/SX PC (Sox-416 Macro, Gerhard, Germany). Liver and intestine sections of 4–7 μm were taken and stained with hematoxylin and eosin. All sections were examined under a microscope (Bresser Science TRM 301, Bresser GmbH, Rhede, Germany) and any histological abnormalities were recorded. A digital camera (Bresser MikroCam 5.0 MP, Bresser GmbH, Rhede, Germany) adjusted to the microscope was used for acquiring histological section images. For feeding trial II, 2 fish per tank were randomly sampled (six fish per dietary group). The same procedures as in the feeding trial I were followed for histological examination.

A semi-quantitative grading system was used in order to quantify the histopathological alterations of the examined tissues [28]. Severity grading used the following system: Grade 0 (not remarkable), Grade 1 (minimal), Grade 2 (mild), Grade 3 (moderate), Grade 4 (severe).

### 2.3. Microbiota Analysis-DNA Extraction, Bioinformatics and Data Analysis

In the present study, the effect of PBM and HFM on the intestinal microbiota of juvenile *S. aurata* was investigated at the 50% FM replacement level that negatively affected the fish growth performance see [7,8]. Thus, for the microbiota analysis fish from PBM50 and HFM50 groups of the feeding trial I were used. Two fish per tank (eight fish per dietary group) were randomly sampled and dissected using sterile lancets and forceps. The midgut was transferred in sterile particle-free (<0.2 mL) sea water (SPFSW). The gut's contents were extruded by mechanical force with forceps, as we targeted the resident gut microorganisms and not the ones associated with the ingested food. DNA extraction and 454 tag-pyrosequencing were performed as shown at Nikouli et al. [29].

Processing of the resulting sequences, i.e., trimming and quality control, was performed with the MOTHUR software (v 1.35.0 open access, University of Michigan, MI, USA) [30]. Only sequences with ≥250 bp and no ambiguous or no homopolymers ≥8 bp were considered for further analysis. These sequences were aligned and classified using the SILVA SSU database (release 119) [31]. All sequences were binned into Operational Taxonomic Units (OTUs) and were clustered (average neighbor algorithm) at 97% sequence similarity. Coverage values were calculated with MOTHUR (v 1.35.0). The batch of sequences

from this study has been submitted in NCBI Short Read Archive under accession number SRS1839183. The heatmaps of the dominant OTUs and orders were implemented by the pheatmap function in the pheatmap package in R version 3.0.2). For the prediction of abundant metabolic pathways the Piphillin algorithm [http://secondgenome.com/Piphillin (accessed on 1 October 2020)] was used with support of KEGG database [32].

### 2.4. Statistical Analysis

For the microbiota analysis, canonical correspondence analysis (CCA) was performed using the R package vegan [33]. Similarly, the significance of morphological parameters and diversity indices for the ordination of the samples was calculated using the function envfit of the same package. Differentially abundant categories (taxa or subsystems) between samples were identified with DESeq package version 1.14.0 [34] using the binomial test and false discovery rate ($p < 0.05$). For liver fat data, percentages were subjected to one-way analysis of variance (ANOVA) followed by Tukey's post-hoc test to rank the groups using SPSS 18.0 (SPSS, Chicago, IL, USA).

### 3. Results

#### 3.1. Liver Histology

In fish fed the control FM diets only minimal alterations (grade 1) were detected in their hepatic tissues (Table 1, Figures 1A and 2A). In general, the liver had normal structure with central hepatocytes' nuclei and a small amount of lipid droplets in their hepatocytes cytoplasm. In some of the hepatocytes the nuclei were not central but pressed against the periphery of the cells. In the cytoplasm of the exocrine pancreas' pyramidal cells many large eosinophilic zymogen granules were observed. Fish fed the diets with low inclusion levels of PBM (PBM25 and PBM25+ diets) showed a similar histomorphology to that of the control FM group (Figure 1B,C) and only two fish of the PBM25+ group showed large lipid droplets around pancreatic islets (Figure 1D). Fish fed the diets with a higher inclusion level of PBM, had mild (PBM50 fish, grade 2, Figure 1E) to moderate (PBM100 fish, grade 3, Figure 1F) alterations. The latter group had also increased signs of degeneration (Figure 1F). In some of the hepatocytes, the nuclei were not central but pressed against the periphery of the cells (Figure 1E). Within the hepatocytes, medium size lipid droplets were observed, but no steatosis or liver hemorrhage signs were detected to any fish. The liver histomorphology of fish fed the PBM50+ diet that was supplemented with lysine and methionine was slightly better than that of PBM50 fish.

**Table 1.** Severity score (0–4) for the observed histopathological alterations and liver fat (% of dry weight) in *S. aurata* fed the experimental diets.

|  | FM | PBM25 | PBM25+ | PBM50 | PBM50+ | PBM100 | HFM25 | HFM25+ | HFM50 | HFM50+ | HFM100 |
|---|---|---|---|---|---|---|---|---|---|---|---|
|  | | | | | Severity score | | | | | | |
| Liver | 1 | 1 | 1 | 2 | 2 | 3 | 1 | 1 | 3 | 2 | 4 |
| Intestine | 0 | 0 | 0 | 0 | 0 | 0 | 0 | 0 | 0 | 0 | 0 |
| Liver fat (%) | 38.0 | 36.2 | 40.6 | 42.0 | 43.2 | 42.5 | 35.7 | 36.9 | 42.7 | 40.8 | 7.8 |

As far as the effect of dietary hydrolyzed feather meal is concerned, fish fed with high inclusion levels of HFM showed moderate (grade 3, HFM50 fish) to severe (grade 4, HFM100 fish) alterations in their liver tissue (Figure 2C,D). These fish showed enlarged lipid droplets, signs of pancreatic islets necrosis and hemorrhage, which were more intense in the HFM100 fish. Moreover, the latter fish showed signs of liver cirrhosis (Figure 2D) with the regenerative nodules of hepatocytes to be surrounded by fibrous connective tissue. The supplementation of lysine and methionine at the HFM50+ diet resulted in less hepatic alterations (grade 2) and a normal hepatic structure compared to the HFM50 fish, but still large lipid droplets and more hepatocytes with no central nuclei were detected in these fish (Figure 2F). However, the replacement of fishmeal by HFM at lower levels (HFM25,

HFM25+) resulted in a normal liver histomorphology that was similar to that of the control FM group (Figure 2B,E).

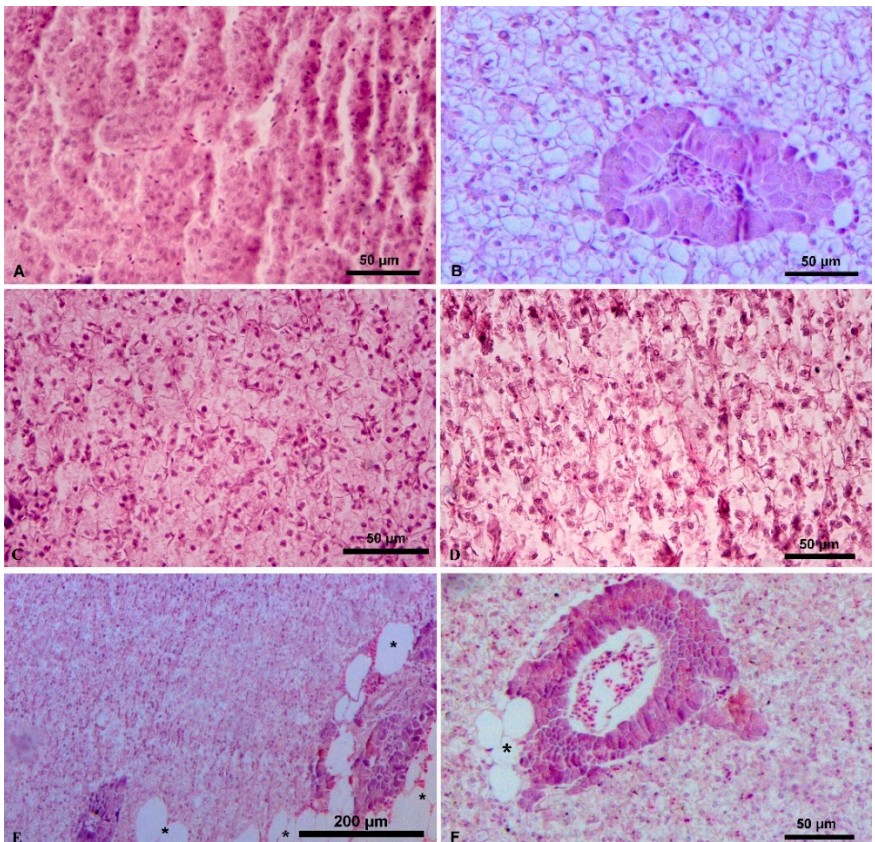

**Figure 1.** Liver histopathological examination of *S. aurata* fed on PBM diets. (**A**) fish fed FM diet—normal liver structure; (**B**) fish fed PBM25 diet—normal liver structure; (**C**) fish fed PBM25+ diet—normal liver structure; (**D**) fish fed PBM25+ diet—large lipid droplets (*) around pancreatic islets in some fish; (**E**) fish fed PBM50 diet—medium size lipid droplets with some nuclei pressed against the periphery of the cells; (**F**) fish fed PBM100 diet—liver degeneration.

### 3.2. Intestinal Histology

All the experimental groups of fish of both feeding trials revealed a normal intestinal histology and none of them showed any signs of inflammation (Figures 3 and 4). Enterocytes were distinct, while goblet cells and apical epithelial vacuoles were normally present (Figure 3A). In addition, abundant eosinophils cells were normally observed within the submucosa layer of all fish (Figure 3B).

### 3.3. Intestinal Microbiota

The gut bacterial diversity of fish was studied using 454-pyrosequencing. From the 24 samples analyzed in total only eight provided a satisfactory number of sequences (>100) combined with coverage >90% (Table 2). Taxonomic and potential species habitat origin was further studied, as well as similarities between the bacterial community composition (BCC) of the different dietary groups. A total of 125 Operational Taxonomic Units (Figure 5A) were identified from all 454 datasets, containing 5876 rRNA sequences in total (Table 2). Coverage was above 95% for all samples, while diversity was low (Shannon < 3) in all samples with the lowest values (<2) being observed in the FM-fed fish (Table 2).

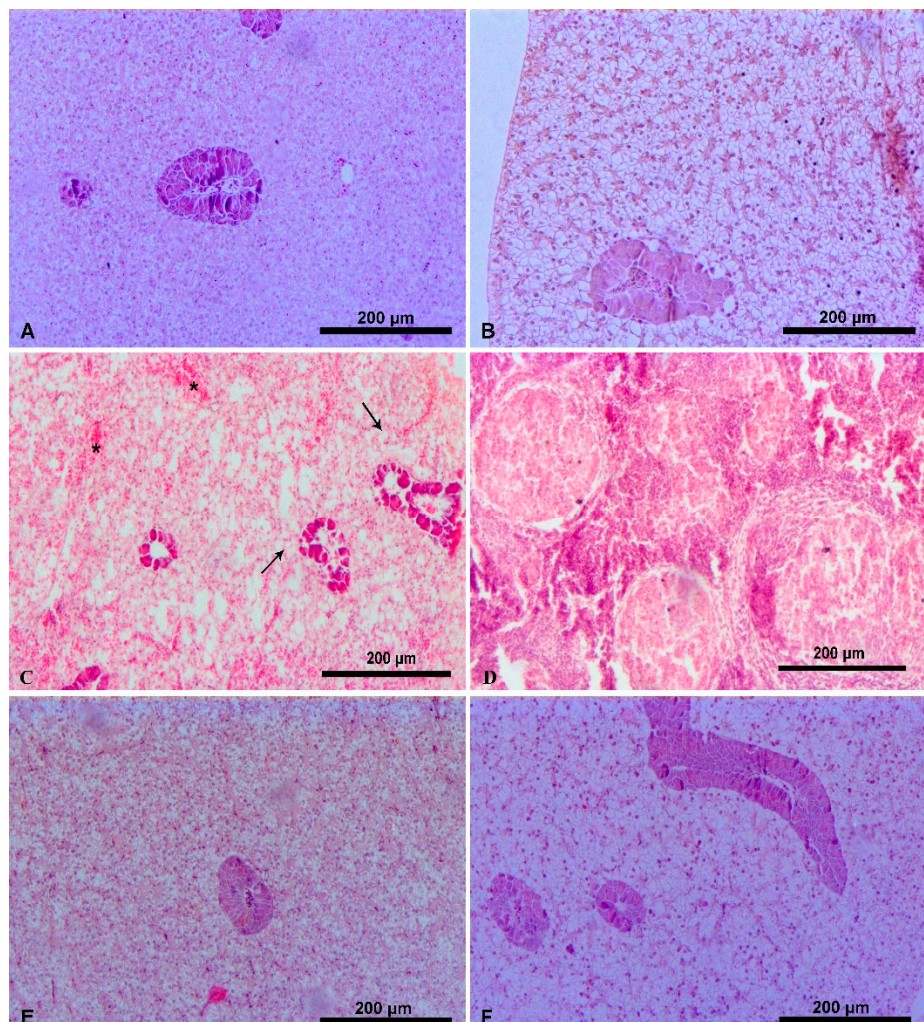

**Figure 2.** Liver histopathological examination of *S. aurata* fed on HFM diets. (**A**) fish fed FM diet—normal liver structure; (**B**) fish fed HFM25 diet—normal liver structure; (**C**) fish fed HFM100 diet—hemorrhage signs (*) and large lipid droplets (arrows); (**D**) fish fed HFM100 diet—signs of liver cirrhosis; (**E**) fish fed HFM25+ diet—normal liver structure; (**F**) fish fed HFM50+ diet—hepatocytes with no central nuclei were detected.

**Table 2.** Sequencing results, diversity indices and coverage values of fish fed the FM, PBM50 and HFM50 diets.

|  | FMa | FMb | HFMa | HFMb | HFMc | HFMd | PBMa | PBMb |
|---|---|---|---|---|---|---|---|---|
| Richness | 14 | 14 | 48 | 29 | 14 | 28 | 15 | 16 |
| Sequences | 2283 | 132 | 2206 | 325 | 196 | 181 | 234 | 319 |
| Shannon | 1.32 | 1.71 | 2.27 | 2.97 | 2.05 | 2.87 | 2.10 | 2.23 |
| Cumulative abundance >1% | 99.21 | 96.21 | 92.84 | 97.54 | 97.45 | 97.24 | 98.72 | 98.12 |
| Coverage | 1.00 | 0.95 | 0.99 | 0.99 | 0.97 | 0.97 | 0.99 | 0.99 |

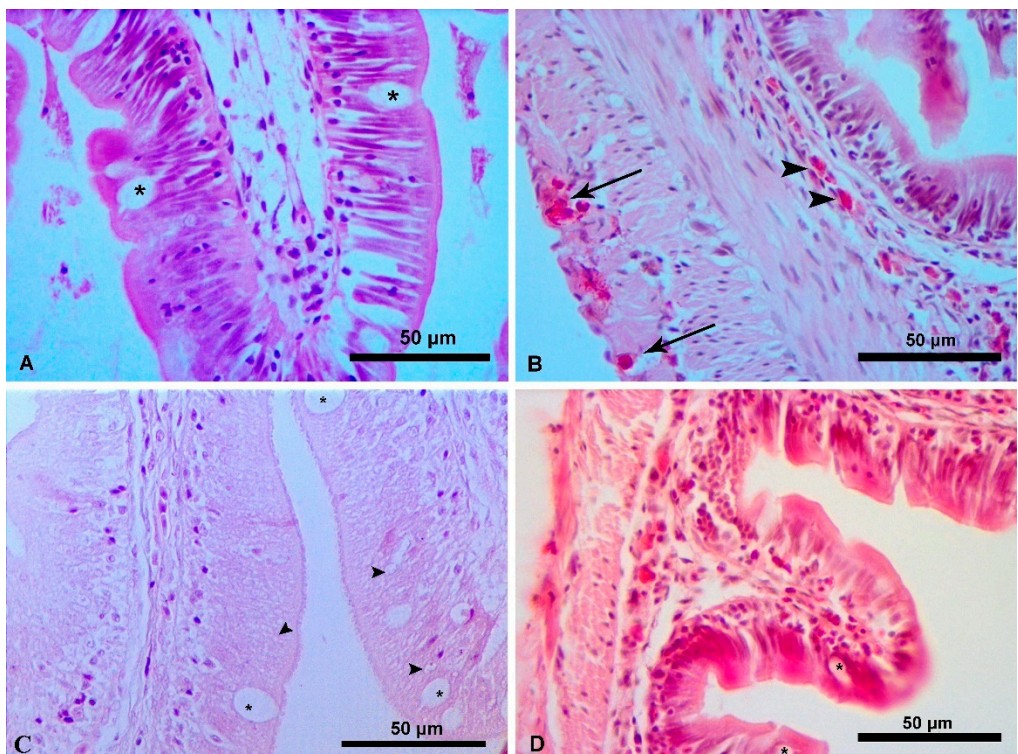

**Figure 3.** Midgut histopathological examination of *S. aurata* fed on PBM diets. (**A**) fish fed FM diet—normal gut structure with goblet cells (*) present; (**B**) fish fed PBM50 diet—eosinophil cells (arrow) accumulation within the muscularis layer. Abundant eosinophils cells were normally observed within the submucosa layer (arrowhead); (**C**) fish fed PBM25 diet—normal structure; (**D**) fish fed PBM100 diet—normal structure with goblet cells (*) present.

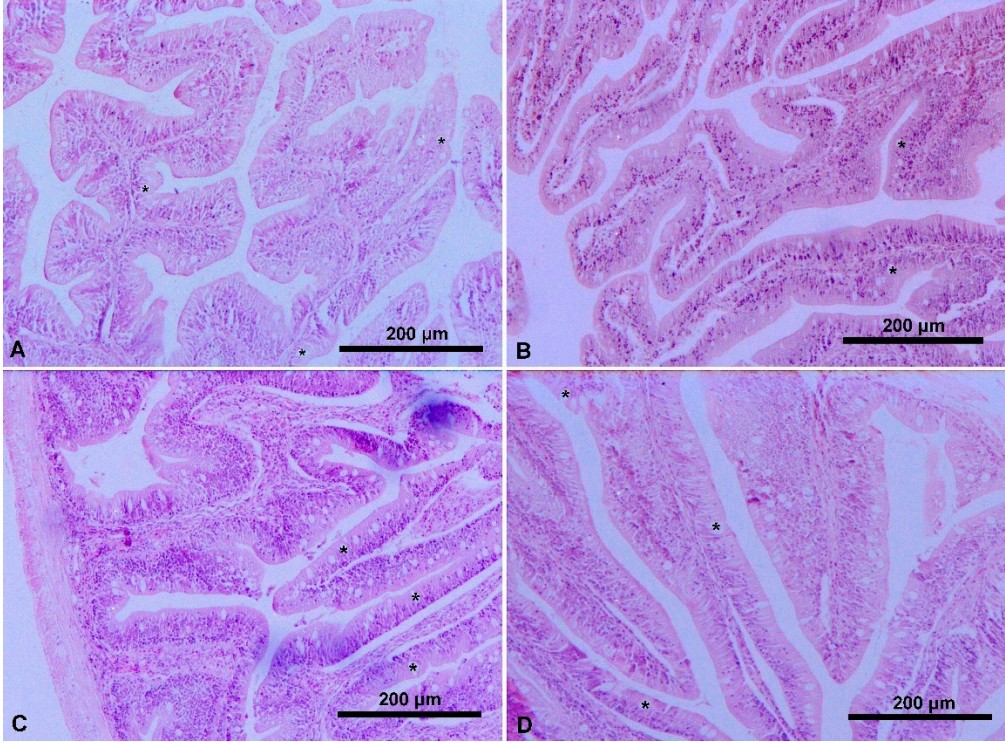

**Figure 4.** Midgut histopathological examination of *S. aurata* fed on HFM diets. (**A**) fish fed FM diet; (**B**) fish fed HFM50 diet; (**C**) fish fed HFM25 diet; (**D**) fish fed HFM50+ diet. In all images, the gut structure appeared normal with goblet cells (*) present.

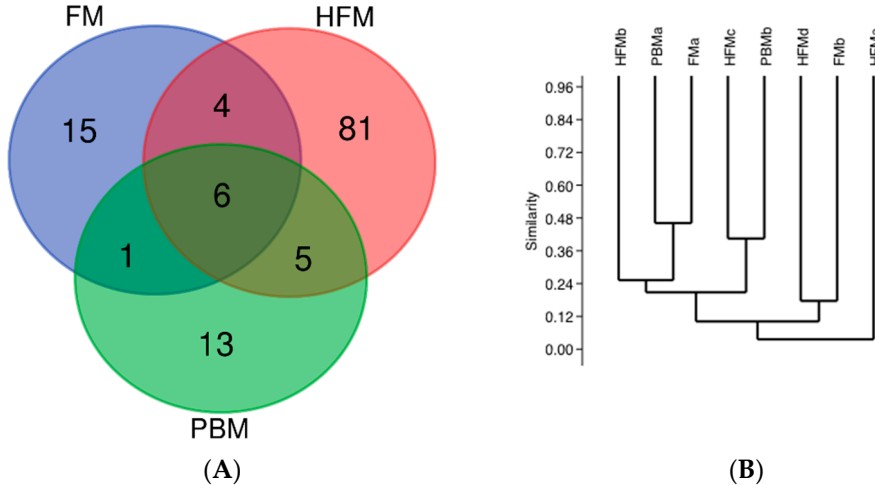

**Figure 5.** (**A**) Venn diagram showing shared and unique OTUs between the dietary groups (FM, PBM50 and HFM50); (**B**) Morisita similarities between fish fed the FM, PBM50 and HFM50 diets.

Morisita similarities between samples were very low (<50%) showing no specific pattern according to the different diet fed (Figure 5B). Canonical correspondence analysis also exhibited no pattern among fish of either the same or of a different dietary group but revealed the importance of body weight for the ordination of the samples ($p < 0.05$) (Figure 6). At the phylum level, all samples were characterized by *Proteobacteria* and *Actinobacteria*, which were the most abundant in almost all fish, as well as by *Bacteroidetes* and *Firmicutes* (Figure 7A). At the OTU level, a total number of 64 OTUs were detected in relative abundances >1% in at least one fish. Overall, these OTUs accounted for more than 97% of the total diversity in all samples (Table 2). Similarly, OTUs with relative abundance >10% clearly representing persistent members of BCC reached cumulative abundances >50% in all samples (Figure 7B) and represented different species of *Alpha-*, *Beta-*, *Gamma-proteobacteria*, *Actinobacteria*, *Flavobacteria*, *Bacilli* and *Clostridia* (Figure 7B). In total, 6 OTUs were shared amongst all dietary groups (Figure 5) and belonged to the genera *Staphylococcus*, *Pseudomonas*, *Delftia*, *Cutibacterium* and *Hydrogenophaga* (Figure 7B). Most of them (5) belonged to the abundant species that dominated (>10%) at least in one sample (Figure 7B). The cumulative abundances of this 'core' microbiome that was identified from habitat ranged from 6.1% (HFMa) to 99.2% (FMa) (Figure 7B). The lowest values for core microbiome relative abundances were observed in the HFM fish with an average of 20.15% contrary to 63.34% and 43.81% in the FM and PBM fish, respectively. This was attributed to the unique abundant species that were detected in the HFM fish, belonging to different *Actinobacteria*, such as *Propiomicromonospora* and to other species, such as *Roseomonas* and *Sphingomonas*.

Differences in predicted functional pathways based on the bacterial abundance did not exhibit any significant grouping of the samples based on the different diet fed. However, some pathways were found to be significantly different ($p < 0.05$) among the different dietary groups (Figure 8). Overall, pathways for renin-angiotensin system, retinol metabolism and cAMP signaling were decreased in fish fed the FM diet compared to those fed either the HFM or the PBM diet. This suggests gut dysbiosis in the two latter groups and possibly an effort to use alternative carbon sources. Steroid degradation pathways showed a statistically significant increase in the HFM fed fish, indicating that microbial communities were using alternative carbon sources.

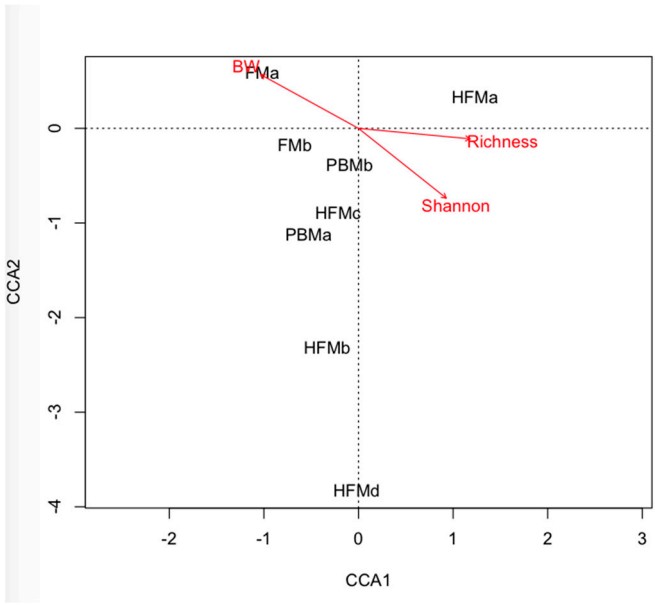

**Figure 6.** Canonical correspondence analysis for fish fed the FM, PBM50 and HFM50 diets. Only factors with significant values ($p < 0.05$) are presented.

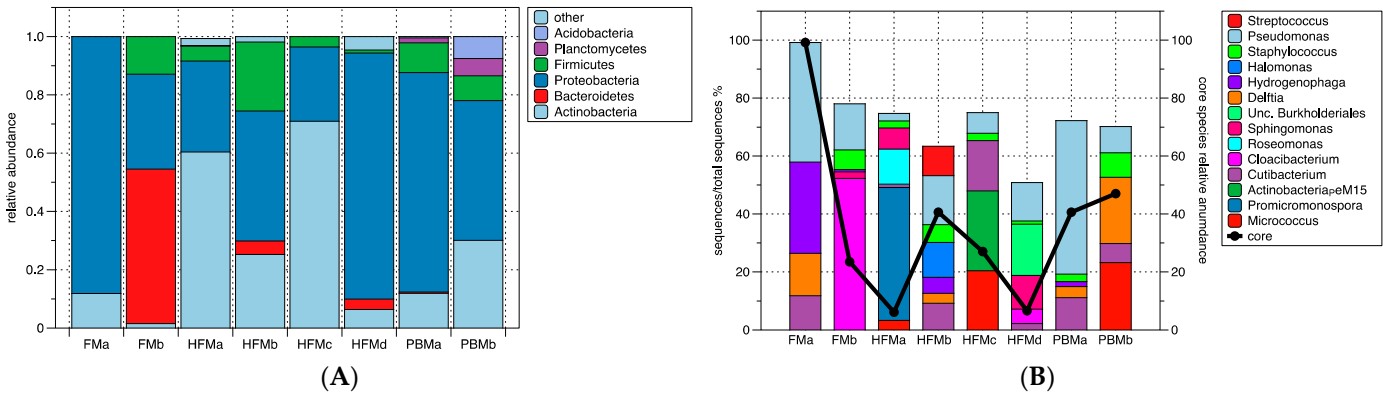

**Figure 7.** (**A**) Relative abundances of different bacterial phyla in fish fed the FM, PBM50 and HFM50 diets; (**B**) Relative abundances of abundant (>10%; left axis) and shared (between treatments) OTUs (right *y*-axis) in fish fed the FM, PBM50 and HFM50 diets.

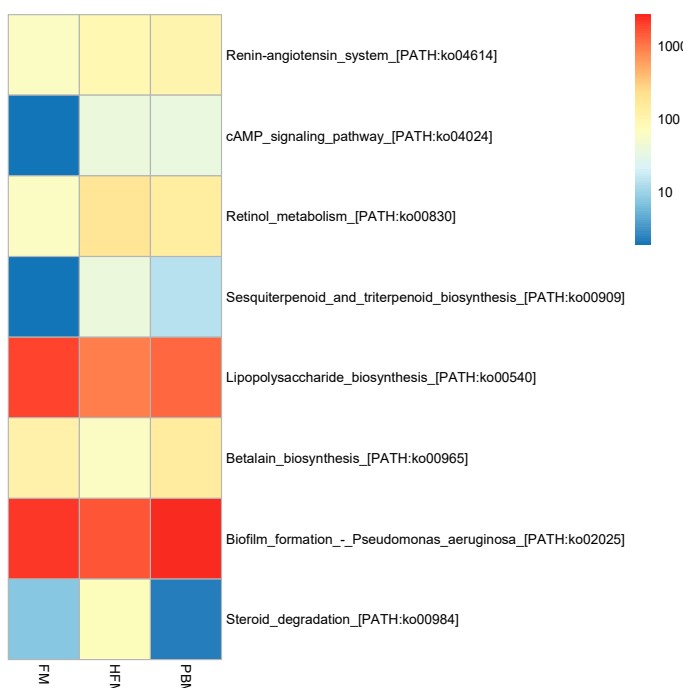

**Figure 8.** Significantly different (*p* < 0.05) predicted functional pathways of the bacterial relative abundances between fish fed the FM, PBM50 and HFM50 diets.

## 4. Discussion

Non-ruminant processed animal proteins, such as hydrolyzed feather meal (HFM) and poultry by-product meal (PBM) have been used successfully to replace fishmeal protein in the diets of several farmed fish and crustacean species [5–8,35–39]. However, knowledge of their effects on the histology of digestive organs, the intestinal microbiota and digestive physiology is extremely limited. Histology is a valuable tool that is used to describe tissue alterations and to detect any possible pathological signs in fish that may be caused by dietary protein modifications. In addition, intestinal microbiota profiling may assess fish intestinal function, health and nutrition [40,41].

### 4.1. Liver Histology

In the present study, the inclusion of poultry by-product meal or hydrolyzed feather meal caused no to severe alterations in the hepatic tissue of *S. aurata* and these alterations were dependent on the level of fishmeal protein replacement. Neither PBM nor HFM altered the liver histomorphology of seabream when these animal proteins replaced fishmeal at 25%. However, at higher replacement levels more lipid droplets and increased hepatic vacuolization were observed, and these changes were more pronounced in fish fed HFM diets. In general, high inclusion levels of PBM caused mild to moderate hepatic alterations compared to the high inclusion levels of HFM that caused severe alterations, particularly in the case of total fishmeal replacement. Although there were no signs of steatosis, which may be caused by the increased lipid vacuolization, the total FM replacement by HFM, contrary to PBM, led to haemorrhage, pancreatic islet necrosis and cirrhosis in a substantial number of fish that were examined. The dietary supplementation of PBM and HFM with essential amino acids, such as lysine and methionine, seemed to improve the digestive physiology. Fish fed these diets showed fewer hepatic alterations and abnormalities compared to fish fed diets of a similar replacement level but without amino acid supplementation.

The present findings contradict with those reported by Sabbagh et al. [42] in which the 100% replacement of FM by PBM did not cause any liver alteration in *S. aurata*. Although in both studies the total FM replacement by PBM did not lead to any clear signs of steatosis, the mild increase in lipid vacuolization with the increase of PBM dietary inclusion observed

in the current study may indicate a lower lipid digestibility of the PBM fat. This indication is supported by the fact that fish fed diets with high inclusion levels of PBM had increased fat in their livers (Table 1), although this cannot be clearly said for the HFM fed fish. It has been suggested that that a hepatic lipid accumulation may occur because the excessive dietary intake of lipids that surpasses the physiological capability of the liver to β-oxidize them, thus leading to larger lipid droplets and subsequent steatosis [43]. However, in our study the dietary lipid intake of PBM and HFM fed fish was lower than that of FM fed fish see [7,8], which indicates that dietary lipids, specifically of PBM and HFM lipids, inducehepatic lipid accumulation in *S. aurata*.

Increased lipid droplets and hepatic vacuolization with signs of inflammation have also been reported in *Lates calcarifer* [22,44] and in *Tinca tinca* [45] when the FM in the fish fed diets were completely replaced with PBM. Moreover, Zhou et al. [21] reported an induced steatosis in the hepatocytes of hybrid grouper even when fish were fed diets with 50–70% FM replacement levels by PBM. On the other hand, the total replacement of FM by PBM did not cause any increased vacuolization or alterations in the hepatocytes of *Salmo salar* [14] and of *Oreochromis niloticus* [46]. The effects of dietary HFM as a sole FM replacement on fish liver histomorphology are not well studied. Hartviksen et al. [14] stated that a diet with total FM replacement by HFM did not reveal signs of steatosis in *Salmo salar* with hepatocytes having even a lower fat accumulation than to those fed on FM. Studies using HFM in a blend with other animal proteins, including PBM, for FM replacement have reported that high substitution levels induce hepatic lipidosis and steatosis in *Lateolabrax japonicus* [19] and in hybrid grouper [20].

### 4.2. Intestinal Histology

The inclusion of poultry by-product meal or hydrolyzed feather meal did not cause any intestinal histological alterations in seabream compared to fish fed the FM-based diet. All fish showed distinct enterocytes with abundant eosinophils cells, with goblet cells and apical epithelial vacuoles being present along the entire intestine of fish. Goblet cells assist fish health and nutrition as the mucus secreted by them acts as a protection medium to the epithelium, while also lubricates undigested materials for onward progression into the rectum [47,48]. Moreover, apical epithelial vacuoles consist integral structural components of the intestine that are responsible for nutrient absorption [49]. Although the present findings do not provide a sufficient evidence for the absorption of PBM and HFM, it can be claimed that these land animal proteins did not result in signs of malnutrition or inflammation, such as enteritis. This applies for all tested FM replacement levels, except for total replacement by HFM which was not feasible to examine. Similarly, an unaffected intestinal histology was reported in Atlantic salmon fed on high levels of PBM [50] and in Nile tilapia fed on high levels of HFM [51] replacing dietary FM. Hartviksen et al. [14] working with Atlantic salmon reported no severe signs of enteritis and unaffected numbers of eosinophilic granular cells in fish fed either with PBM or HFM replacing dietary fishmeal at a level close to 50%. However, the authors reported that PBM led to a decreased submucosa width, while HFM led to a decreased presence of goblet cells and an increased presence of apical epithelial vacuoles in the intestine. Moreover, Chaklader et al. [22] reported a dysregulated intestinal morphology with smaller microvilli of shorter diameter in juvenile barramundi fed on PBM totally replacing dietary fishmeal. Furthermore, Yu et al. [23] working with Pengze crucian carp reported shortened microvillus and enterocytes and thinner muscular thickness when HFM replaced more than 30% of dietary fishmeal protein.

### 4.3. Intestinal Microbiota

Regarding the dietary effects on the intestinal microbiota, it can be argued that no clear effects were detected from the use of HFM or PBM. The gut Bacterial Community Composition (BCC) of *S. aurata* fed either FM, PBM or HFM was characterized by groups that are commonly found in the fish gut microbiome. The core genera *Pseudomonas*, *Cutibacterium*, *Staphylococcus* and *Delftia* have been previously reported as core microbiome

of *S. aurata* farmed in several geographical sites [29,52], suggesting that certain bacterial genera are capable of colonizing the seabream gut independently of the diet and location. Similar findings have been reported for *Salmo salar* [41] with authors stating that the role of fish-hosts in selecting or promoting core microbes is unclear. *Actinobacteria* species were dominant and unique in the HFM fed fish and their dominance could be related to their antimicrobial functions (i.e., antibiotics) that protect the host [53,54]. Additionally, the prevalence of terpenoid biosynthesis genes that were prevalent in the HFM fed fish is mostly detected in *Actinobacteria* [55] and was in accordance with the taxonomic BCC (Figure 7A,B). The *Sphingomonas* species in the gut have been found to be negatively correlated with weight gain [56] which also agrees with our CCA data (Figure 6).

The predicted pathways that were enriched in the HFM and PBM fed fish compared to FM fed fish were mostly related to functions that imply gut dysbiosis. For instance, increased renin-angiotensin system (RAS) genes have been related to RAS system activation which implies malnutrition [57,58]. Gut dysbiosis that promotes RAS activation is mostly related to decreased abundances of fermenting bacteria, which was the case in the HFM and PBM groups of fish. Similarly, bacterial retinol metabolism predicted genes indicate a potential need for vitamin A production that enhances mucosal immunity. Thus, bacteria able to participate in retinol metabolism also assist in avoiding pathogen invasion [59].

Knowledge of the effects of fishmeal replacement by land animal proteins on the intestinal microbiota is extremely limited. In the present study, as stated above, there was no specific pattern in the bacterial communities among fish fed either fishmeal or the tested land animal proteins. Gajardo et al. [41], working with *Salmo salar*, reported a significant effect of the dietary PBM on the distal intestine digesta and mucosa. This study found significantly higher and lower abundances of specific genera in fish fed PBM than in fish fed on a FM-based diet. Hartviksen et al. [14], also working with *Salmo salar*, reported that the use of HFM and PBM as fishmeal replacements increased the total allochthonous and total autochthonous bacteria in the distal intestine, but the total autochthonous bacteria in the proximal intestine remained unaffected. The authors also reported that the supplementation of HFM caused significant increases in specific genera (*Corynebacteriaceae*, *Lactobacillaceae*, *Streptococcaceae*, *Pseudomonadaceae*, *Xanthomonadaceae*) and decreases in another (*Vibrionaceae*). Furthermore, PBM caused increases in *Corynebacteriaceae* and decreases in b-*Proteobacteria*, *Bacilli-like*, *Peptostreptococcaceae* and *Vibrionaceae*. Certainly, a better understanding of the functional roles of the intestinal microbiota communities of fish and to what extent these are affected by the use of land animal proteins is needed.

## 5. Conclusions

In conclusion, fishmeal replacement by either poultry by-product meal or hydrolyzed feather meal did not cause any intestinal histological alterations. Thus, these results indicate normal digestion and absorption in the midgut of *S. aurata* even when their dietary fishmeal protein is completely replaced. Neither land animal proteins altered the liver histomorphology of gilthead seabream when fishmeal was replaced at 25%. However, at higher replacement levels increased lipid droplets and hepatic vacuolization were observed to be more pronounced in fish fed HFM diets. Moreover, the dietary supplementation of PBM and HFM with essential lysine and methionine seemed to improve the digestive physiology, as fish fed these diets showed fewer hepatic alterations and abnormalities compared to diets of a similar replacement level but without amino acid supplementation. The dominant phyla in the intestinal microbiota were *Proteobacteria* (58.8%) and *Actinobacteria* (32.4%), but no specific pattern was observed among the different dietary fish groups at any taxonomic level. This finding was probably driven by the high inter-individual variability observed.

**Author Contributions:** Conceptualization, I.T.K., E.M. and K.A.K.; Methodology, P.P., A.M. and P.B.; Software, P.P., A.M. and P.B.; Validation, I.T.K., E.M. and K.A.K.; Formal Analysis, P.P., A.M. and P.B.; Investigation, P.P., A.M. and P.B.; Resources, I.T.K., E.M., P.B. and K.A.K.; Data Curation, I.T.K., K.A.K. and E.M.; Writing—Original Draft Preparation, P.P. and A.M.; Writing—Review & Editing,

I.T.K., P.B., K.A.K., E.M.; Supervision, I.T.K., E.M. and K.A.K.; Project Administration, I.T.K.; Funding Acquisition, I.T.K. All authors have read and agreed to the published version of the manuscript.

**Funding:** This research was funded by the Operational Programme "Fisheries 2007–2013" (Ministry of Rural Development and Food of the Hellenic Republic/European Fisheries Fund) under the project title "The use of Processed Animal Proteins in the feeds of gilt-head seabream (*Sparus aurata*)", Grant Number: 2014ΣE086800090000.

**Institutional Review Board Statement:** The study was conducted according to the guidelines of EU legal frameworks related to the welfare and protection of animals for scientific purposes (Directive 2010/63/EU) and in strict accordance with the University of Thessaly's Ethics Committee on the Use of Animals in Research (protocol approval 20 December 2016).

**Informed Consent Statement:** Not applicable.

**Data Availability Statement:** The data presented in this study are available on request from the corresponding author.

**Conflicts of Interest:** The authors declare no conflict of interest.

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
