# Peer review of "Effects of Dietary Fishmeal Replacement by Poultry By-Product Meal and Hydrolyzed Feather Meal on Liver and Intestinal Histomorphology and on Intestinal Microbiota of Gilthead Seabream (Sparus aurata)"

_applsci, doi:10.3390/app11198806_

Round 1

Reviewer 1 Report

This manuscript is a study about the effects of dietary replacement of fishmeal with by-product meal and hydrolyzed feather meal on the histomorphology of liver and intestine, and intestinal microflora of gilthead seabream. The work is clearly presented, ant the manuscript has an interesting scientific message, and it might be of interest to some readers. However, several major issues need to be addressed. The English language of the manuscript needs to be improved, and there are many grammatical mistakes.

  1. Introduction

-L35-37: This sentence needs to be rewritten.

-L58: It is suggested to briefly introduce the research progress of by-product meal and hydrolyzed feather powder replacing fishmeal in the growth, digestion and immunity of aquatic animals.

-L75: “Sparus aurata” be in italics.

  1. Materials and Methods

- The details or references about the feed formula should be provided.

- Data about the growth performance in Table 1 is overlapped with the data from the authors' published papers (Karapanagiotidis, et al, Aquaculture Nutrition, 2019, 25, 3-14. 440; Psofakis, et al, Aquaculture 2020, 521, 735006). It is suggested to delete this table from this manuscript, and the authors could briefly introduce the previous findings in the introduction section.

  1. Results

- It is recommended that the magnification of images be unified, such as Figure 1, and the typical features can be partially magnified in the corresponding subfigures.

- It is suggested to present the semi-quantitative grades of the liver histology in a table, which would be more intuitively to display the results.

- As stated in the manuscript, intestine is the primary site of nutrient absorption, and the intestine was too thin in the fish of HFM100 group. Hence, I suggest the authors to measure and count the villus height and intestinal wall thickness in different groups. See references as follows:

Effects of dietary glucose and sodium chloride on intestinal glucose absorption of common carp (Cyprinus carpio L.), Biochemical and Biophysical Research Communications, 495, 1948-1955.

Intestinal morpho-physiology and innate immune status of European sea bass (Dicentrarchus labrax) in response to diets including a blend of two marine microalgae, Tisochrysis lutea and Tetraselmis suecica, Aquaculture, 2019, 500, 660-669.

  1. Discussion

- It is suggested that the authors should reorganize the discussion section in the order of the results section.

-L294-298: the authors examined the histological changes only in the midgut. How can it be concluded that poultry by-product meal is normally digested and absorbed in the distal intestine? Histological changes are not sufficient evidence for the absorption of poultry by-product meal and hydrolyzed feather meal.

  1. Conclusions

-L394: distal or mid intestine? midgut was sampled in this study.

Reviewer 2 Report

This Ms addresses an important topic  of a very important important cultured  species as Sparus aurata. The Ms is properly organized and each section is clearly and adequately presented. One minor suggestion refers to the inclusion in the key-words the name of species considered in this investigation: Sparus aurata.

As mentioned above, the topic is well addressed in the Introduction, the relevant questions are includes, the Methods are adequate and sufficiently described and the same applies to the other sections of the Ms.

Under my personal point of view, using poultry-based components for feeding fish species will contribute to increase the pressure on an already heavy production sector with the concomitant ecological concerns that this might imply. Maybe the authors could address this topic in the Discussion section.

Another topic that is not completely clear to me are the eventual cumulative effects of using the contribution of poultry by-product meal and hydrolyzed feather meal beyond the 100-day period used in this investigation. I think that a 100-day period is not enough to produce a marketable-sized sea bream and, if considered relevant, this comment would imply extending the length of the experiments or, alternatively, suggest to the authors include eventual data that might be available.

Author Response

We would like to thank the reviewer for her/his thoughtful comments and constructive suggestions, which helped us to improve the quality of the manuscript. 

The followings are our point-by-point responses to reviewers’ revisions.

  • Inclusion of key-word. Response: In the revised version, we have included in the key-words the name of species Sparus aurata.
  • using the contribution of poultry by-product meal and hydrolyzed feather meal beyond the 100-day period. Response: Indeed, it would be of much interest to see the effects on a longer period and at the marketable size fish. However, as it is commonly accepted, the vast majority of the feeding trials are conducted for 6-10 weeks that is much shorter period from that we used here (14-15 weeks). This is because the dietary effect is fast and direct. Therefore, we believe that our findings are reliable for the time examined. Moreover, the vast majority of the feeding trials are not conducted at the marketable size but rather at juvenile stage due to technical and costly limitations.
  • poultry-based components for feeding fish species will contribute to increase the pressure on an already heavy production sector. Response: we are not sure what the reviewer means here. Indeed, the use of poultry meal in aquafeeds also has an environemntal impact, but this is considered much lower than that of fishmeal.